# Walking within a Crowd Full of Virtual Characters: Effects of Virtual Character Appearance on Human Movement Behavior

Category: Research

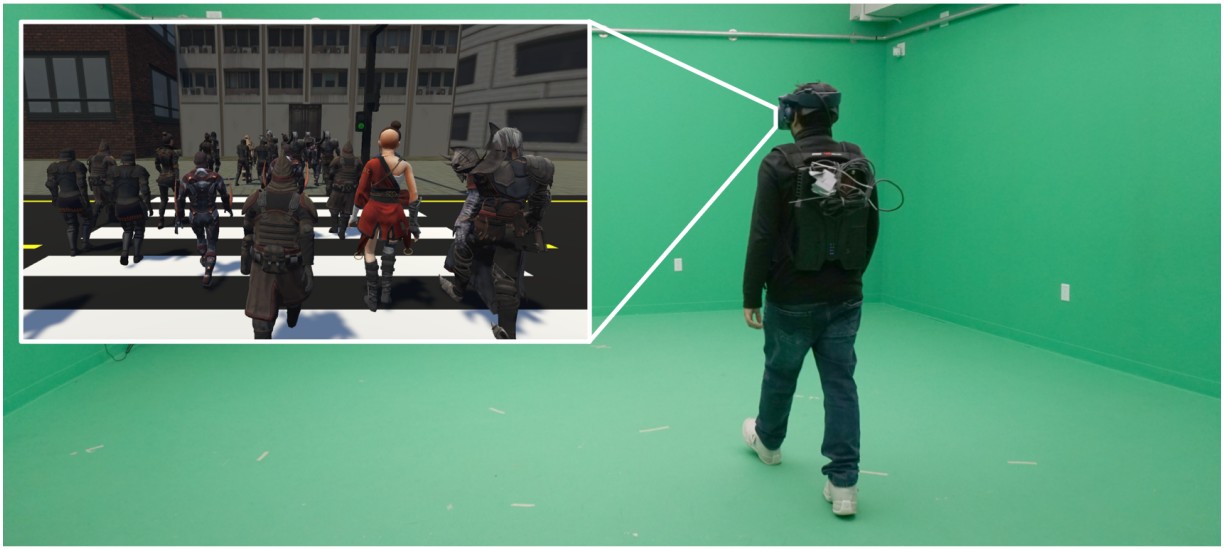

Figure 1: A participant walking in the motion capture studio and moving toward the opposite sidewalk in the virtual metropolitan city, and a third-person view of the virtual environment and virtual crowd the participant was observing. The participant is wearing all of the devices used in this study (an MSI VR One backpack computer, and an HTC Vive head-mounted display).

## ABSTRACT

This paper is a study on the effects that a virtual crowd composed of virtual characters with different appearance has on human motion in a virtual environment. The study examines five virtual crowd conditions that include the following virtual characters: neutral, realistic, cartoon, zombies, and fantasy virtual characters. Participants were instructed to cross a virtual crosswalk and each time, one of the examined crowd conditions shown. The movement behavior of participants was captured and objectively analyzed based on four measurements (speed, deviation, trajectory length, and interpersonal distance). From the results, it was found that the appearance of the virtual characters significantly affected the movement behavior of participants. Specifically, participants walked slower when exposed to a virtual crowd composed of cartoon characters and faster when exposed to fantasy characters. It was also found participants deviated more when exposed to a crowd composed of fantasy characters compared to a crowd composed of cartoon and zombie characters. Finally, the interpersonal distance between participants and fantasy characters was significantly greater compared to human and zombie virtual characters. Our findings, limitations and future directions are discussed in the paper.

**Index Terms:** Human-centered computing—Human computer interaction (HCI)—Interaction paradigms—Virtual reality; Human-centered computing—Human computer interaction (HCI)—HCI design and evaluation methods—User studies

## 1 INTRODUCTION

In our modern age, navigating crowds are an unavoidable aspect of participating in society. In order to travel any significant distance, it is nearly impossible to not end up as part of one. They are found on our sidewalks, in shopping centers, and any place that humans congregate. It naturally follows then that they will be found in our virtual societies as well. From films and games to interactive simulations crowd forms and provides a sense of realism for these virtual spaces. Although there is significant research on the analysis, modeling, and simulation of crowds and crowd dynamics [10, 15, 25, 32, 35] as well as a number of published papers have examined interactions with virtual characters or groups of virtual characters, only a limited number of studies have examined the way that humans walk along with virtual crowd populations surrounding them [4, 19, 23, 30]. However, there are no conclusive results as to whether a moving virtual crowd population affects the movement behavior of humans or as to how it would have such an effect [4, 23]. In turn, understanding how humans perceive and interact with these simulations of society may prove to be vital for effectively guiding humans through them.

An overlooked factor of crowd simulations is the composition of the crowd. Generally, a majority of the crowd and interaction research inside virtual reality seems to employ characters that share fairly similar characteristics to human beings i.e. it is a normal human crowd. Due to the aforementioned limitations, an understanding of how the appearance of virtual characters that belong to a moving virtual crowd might or might not affect the movement behavior of humans could be quite beneficial for virtual reality developers since it would allow them to develop the parts of virtual reality experiences that include immersive interaction with moving virtual crowds more effectively and precisely.

This study explores human movement behavior in immersive virtual crowds and tries to answer whether and how the appearance of virtual characters that belong to a virtual crowd affected the movement of participants that have been simply instructed to cross a virtual crosswalk and reach the opposite sidewalk (see Figure 1). In this study, based on the appearance of the virtual characters, five crowd conditions namely neutral, realistic, cartoon, zombies, and fantasy were examined. Participants were placed into a virtual

metropolitan city and were instructed to cross a virtual crosswalk and reach the opposite sidewalk while surrounded by a virtual crowd population that was moving toward the same direction. During the walking task, the movement behavior of participants was captured and four measurements (speed, deviation, trajectory length, and interpersonal distance) were extracted. Considering that researchers determined that as the characters started deviating more towards non-human-like characteristics, the eeriness factor - i.e. the inhuman and otherworldly characteristics of the character - increased with them [6], this paper tries to understand the impact that a crowd has upon a human, inside virtual reality environments when the composition changes from neutral to pleasant and aversive appearance of the virtual characters that compose the virtual crowd.

The structure of this paper is organized as follows. In Section 2 related work is discussed. Section 3 covers the methodology of the experimental study. Section 4 presents the results. A discussion of the results is drawn in Section 5. Finally, Section 6 covers the conclusions are future research directions.

## 2 RELATED WORK

In this section, we present related work on how virtual characters have been studied and how humans interact differently with different virtual characters. We also explore the previous work regarding crowd interactions in virtual reality environments and the realism of different crowds. We then move to research regarding the approach-ability or eeriness of different characters in virtual environments.

Numerous studies have been conducted in the past concerning interaction with virtual characters. It has been found that when humans interact with characters inside a virtual environment, the exhibited interaction behavior was similar to that of a situation in real life [26, 28] even if humans are aware that the interaction was taking place with virtual characters and not with real humans [11]. Apart from studies concerning interactions with a single character at a time [26, 33], studies concerning interaction in either small groups of characters or virtual crowds have also been a point of interest to the research community [19].

There has also been published work on how humans interact differently with different virtual characters. de Borst et al. [6] explores emotional elicitation, recognition, and familiarity of virtual characters ranging from human-like to non-human-like. They show that human-like characteristics create a better sense of familiarity and less eeriness. They also show that non-human characteristics, when animated with anthropomorphic expressions, they were also more familiar and approachable. Pan et al. [27] explored how participants mimic a virtual character. They asked participants to follow actions performed by a character and a ball and they refer to this as "anatomical congruency." They found that where participants had to follow a character, their error rate was incredibly low in comparison to where they had to follow a ball. Hamilton et al. [14] focused on how participants mimicked virtual characters. They had asked participants to interact with a virtual character, the virtual character would point at target with varying height and they showed that participants tended to replicate those actions by measuring the elevations of their hands in comparison to the virtual character, which heavily correlated. Latoschik et al. [20] explored the approachability of two virtual characters: humans and mannequins. Participants were asked to perform similar tasks with a human character and a mannequin in a comparative study. They were then asked to rate the attractiveness, eeriness, and humanness of these characters. They showed that humans were far more approachable and less eerie than their mannequin counterparts.

There has been much research done on crowd interactions and walking simulations within virtual environments. A previously conducted study [23] concerning human movement has focussed on the density aspect of the crowd, inside the virtual characters that compose that crowd. The results that were obtained, showed indi-

cations that the participant's movements changed as the density of the crowd as well as the speed increased. Another aspect of crowd simulation is to understand the flow of the crowd. There is a huge implication of how a crowd moves with respect to the participant. Studies showed that crowd movement and outflow was significantly better when the elements in the crowd merged into each other in or under a T-junction, instead of approaching each other from the opposite direction [29].

The flocking behavior participants exhibit while walking with virtual crowds was explored by Moussaïd et al. [22]. They measure the variance in walking speed, shoulder movements, and acceleration. They had two scenarios, one was with a single person in a corridor with an approaching character, and the second scenario was with a crowd that was moving in a single direction and had to exit through a door of varying widths. They showed that there occur patterns while moving which could be attributed to the participants following their closest neighbors while avoiding collisions with other characters. Bruneau et al. [4] explored interactions in regards to crowd sizes in virtual environments. They showed that participants tend to go around small crowds and participants tend to go through larger crowds. Kyriakou et al. [19] explored collision avoidance in virtual crowds and how that relates to realism. They had two scenarios where the participant had to follow a virtual character within a crowd. In the first scenario, the virtual character avoided the crowd, and in the second it did not. They showed that, when participants visualized collision avoidance in crowds, they believed the environment to be more realistic.

The realism of crowds in a goal centric environment also triggered the interest of the research community. In Ahn et al. [1] participants were put in two situations, one where they had a goal to accomplish and one where they did not. They showed that in situations where they had goals to accomplish, participants tend to focus less on the minor imperfections of the crowd, hence those crowds seem more realistic. Ennis et al. [8] explored the realism of synthetic crowds, they showed that participants perceive the realism of a synthetic crowd when the synthetic crowd's positions were based on the positions within a real crowd.

Dickinson et al. [7] showed how crowd densities affect participants' ability to complete a goal. Participants were asked to move objects between tables in a virtual environment. Crowds with varying densities were present within these tables. They showed that crowds with larger densities hindered the participant's movement more. Cirio et al. [5] compared participant interactions within a virtual environment and a real environment using a goal-centric study. Participants were asked to follow a certain target while facing another target. They found that the shape of the trajectories the participants took, their velocity and angular velocity showed similar trends between virtual environments and real life. Sohre et al. [31] explored the role of collision avoidance between virtual character and the participants on overall comfort and perceptual experience in an immersive virtual environment. They found that when collision avoidance was used participants took more direct paths, with less jittering or backtracking, and found the resulting simulated motion to be less intimidating, more realistic, and more comfortable. Finally, Bian et al. [3] created a paradigm of "virtual scenario simulation" to investigate the impact of virtual characters on social behavior in different social interaction contexts. The conducted experiments indicated that both a situational factor (social interaction contexts) and an individual factor (shyness level) could affect the occurrence of the Proteus effect. Moreover, they found that Proteus effects were moderated by the social interaction contexts as well as that in the maintaining interaction context, the Proteus effects were moderated by the level of shyness.

Torre et al. [34] explored how facial expression related to the approachability of characters. In that, tested participants against varying facial expressions and voice tones and showed that consis-

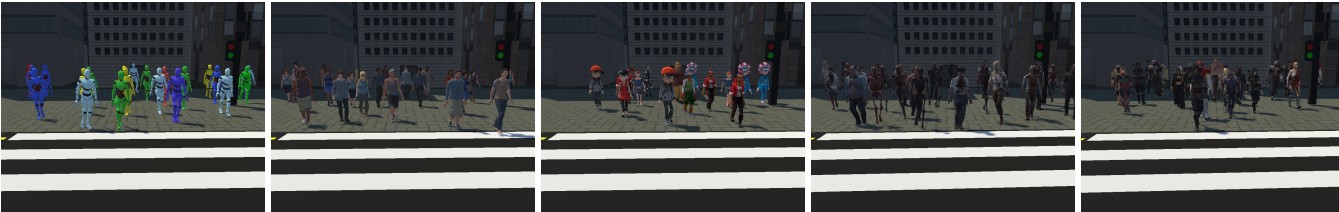

Figure 2: The five different virtual crowd conditions based on the characters' appearance that were used in this study in which participants were asked to walk with and cross a virtual crosswalk. From left to right: neutral crowd (NC), human crowd (HC), cartoon crowd (CC), zombie character (ZC), and fantasy crowd (FC).

tency between facial expressions and voice tone made the characters approachable. Sylaiou et al. [33] compared three different lecture styles in a virtual museum. Lectures were in three different tones: Monotonous, Enthusiastic and Dramatic/Impactful. Participants were then asked to rate the emotion they felt the most. Monotonous elicited an "indifferent" reaction, Enthusiastic elicited an "interested" reaction and Dramatic/Impactful elicited reactions of "shame, anger, and relief." Bailenson et al. [2] explored interpersonal distance for characters in a virtual environment. They showed that when characters approached a participant from the front versus when they approach them from their side, participants tend to be more apprehensive. They showed that participants are more comfortable when characters approach from their peripheral rather than directly in front of them.

There is research that contributes to crowd interactions, virtual character interactions, and the eeriness of characters in virtual environments. Our research leverages elements from these fields and ponders how virtual crowd interactions are influenced by character appearances. There has not been much research done in this field, therefore, our study tries to extend current knowledge and contributes toward this direction.

## 3 MATERIALS AND METHODS

Details on the materials and methods of this study are provided in the following subsections.

### 3.1 Participants

In total, 18 participants, aged 19 to 30 (M=24.89, SD=6.58) took part in our study. Participants were graduate students and faculty of our department recruited by email and class announcements. All students had some experience of being in a virtual environment prior to this study. No participant complained of motion sickness. All participants were volunteers and there was no type of compensation involved. Regarding the sample size (N=18) we would like to note that our decision regarding the small sample is based on a number of similar studies that also investigate movement behavior in virtual environments [4, 5, 17, 23]. It is common in such studies for the sample size to be decreased when the participant trials for a given task are increased.

### 3.2 Experimental Conditions

The study utilized five experimental conditions (see Figure 2) to investigate the effects of a virtual crowd composed of characters with a different appearance on human movement behavior. A within-group study design was implemented to ensure direct comparisons across the experimental conditions. Apart from the virtual character, all other aspects of the virtual environment were identical. The experimental conditions were the following:

- **Neutral Crowd (NC):** This crowd condition includes virtual mannequin characters. The characters are faceless and colored in light blue and red.

- **Human Crowd (HC):** The characters in the crowd are realistic human appearing characters.

- **Cartoon Crowd (CC):** The characters in the crowd are cartoon styled human appearing characters.

- **Zombie Crowd (ZC):** The characters in the crowd are realistic skeletal and clothed zombies.

- **Fantasy Crowd (FC):** The characters in the crowd are realistic fantasy humans.

### 3.3 Setup and Virtual Reality Application

The research was performed at the departmental motion capture studio. The dimensions of the studio were eight meters long and eight meters wide, with a ceiling height of four meters. This studio was appropriate for the experimental study as there were no obstacles, other than a computer desk and a couple of chairs, in the physical world. Therefore, participants were able to walk freely within that studio space.

The HTC Vive Pro head-mounted display device was used for projecting the virtual reality content and the MSI VR One backpack computer (Intel Core i7, NVIDIA GeForce GTX1070, 16GB RAM) was used to run the application. A user wearing the mentioned devices when walking within our motion capture studio space is shown in Figure 1. A virtual reality backpack computer was used for two reasons. Firstly, using cables that connect the head-mounted display to the computer was avoided, as the cables might have affected the movement behavior of the participants. Secondly, the use of a wireless transmitter for the virtual reality headset was avoided as such a transmitter produces latency and therefore this latency might have caused nausea and might have altered the locomotive behavior of the participants. Thus, the use of a virtual reality backpack computer ensured that participants would be able to walk properly in the virtual environment and that the virtual reality content would be transmitted at the proper frame rate.

The application used for this study was developed in the Unity3D game engine version 2019.1.4. A virtual metropolitan city was designed in 3ds Studio Max and then imported to the Unity3D game engine to be used for the study. The virtual environment (crosswalk) used for this experiment is illustrated in Figure 3. The participant is placed on the sidewalk at a crosswalk in the virtual metropolitan city. Virtual pedestrians (virtual crowd) were pre-scripted to cross the road and reach the opposing sidewalk. The virtual characters for each of the conditions were either designed in Adobe Fuse or taken from various Unity Asset Store assets and the animations were assigned using the Adobe Mixamo. Note that all virtual characters had the same height and shoulder width. Each of the characters' crossing scenarios was repeated multiple times. Each character was initialized to surround the participant and was scripted to reach a target position on the opposite sidewalk. After reaching the assigned target position, each character was scripted to move to another location in the virtual environment to help alleviate congestion on the sidewalk. The

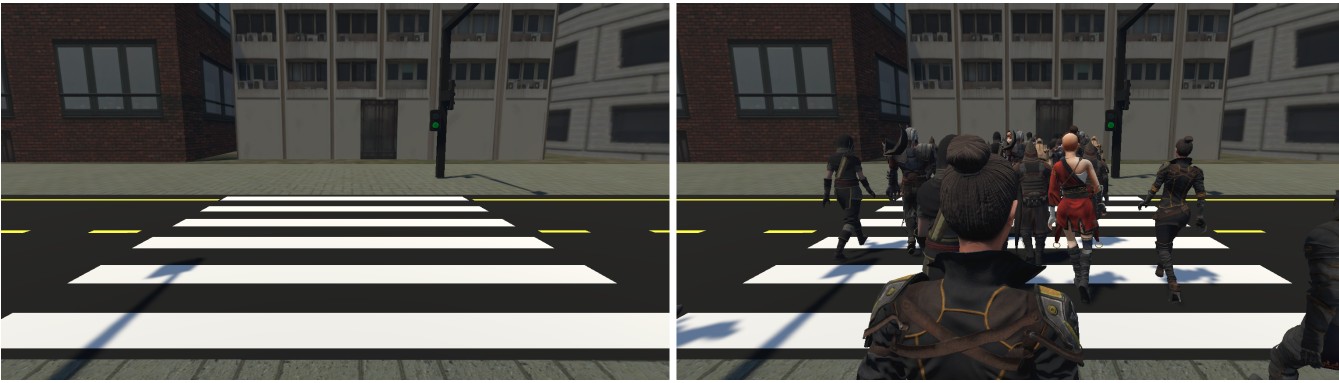

Figure 3: The virtual crosswalk in the metropolitan city (left) populated with a virtual crowd composed by fantasy virtual characters (right). Participants were instructed to cross the virtual crosswalk and reach the opposite sidewalk while being surrounded by the virtual crowd population that was moving toward the same direction.

crowd density, speed, and trajectories were automatically generated by randomly placing the virtual characters in positions that created consistent average movement speed and with a low density. This was achieved using the Unity3D navmesh system and directing navmesh agents towards to their assigned position.

Here, we would like to provide a few of the details regarding the developed virtual crowd. Firstly, each virtual character that belongs to that crowd was implemented to not violate the close phase of the personal space (76 cm) of any other character, according to the proxemics model [13, 16]. Moreover, we set the crowd's speed to not exceed 1.2 m/s. This choice was based on the U.S. Manual of Uniform Traffic Control Devices [9], which defines that the normal walking speed of humans has been estimated to be 1.2 m/s. Finally, we considered the crowd model of Still [32], and more specifically the low-density crowd that places one pedestrian per square meter; therefore participants exposed within a virtual crowd that gave the ability to move freely. These three parameters were constant across conditions, and we consider them as key aspects that helped us standardize the experimental conditions. Finally, we would like to note that the research team decided not to represent the participants with a self-avatar. One previous study found that representing participants with virtual avatars during a walking task might affect their movement within a virtual environment [21]. This was important to consider since the omission of an avatar to represent participants helped in extracting movement behavior that was not influenced by parameters other than those examined, such as a virtual body that does not match the participant's own body in size and appearance.

### 3.4 Measurements

The objective measures are a combination of factors that have been identified by our review of the previous literature. Each of the measures is taken in a series of one hundred in set intervals along the path walked by the participants. The speed and deviation are measures that interpret the general movement behavior that participants take. Speed would tell us if the participants either felt more at ease or on edge in their walk, while the deviation would show that the participants drifted in the crowd. The length of the trajectory and the average interpersonal distance are measures that provide insight on any avoidance behaviors that might occur during the walk. If significant, the length of the trajectory will show that participants deviated during the walk to avoid the characters in the crowd. The interpersonal distance being significantly different would show that the participants tried to keep a distance from the characters without factoring in a changing trajectory. The examined measurements were as follows:

- **Speed:** The average speed of the participants' walking motion from the start to the goal position. The speed was measured in meters/second.

- **Length:** The total length of the captured trajectory (distance traveled) when crossing the road. This was measured in meters.

- **Deviation:** The average deviation (absolute value) between the global trajectory of the virtual crowd and the trajectory of the participant. The average deviation was measured in meters.

- **Interpersonal Distance:** The average distance of the closet four characters directly in front of the participant. The chosen four characters were the same for the participants and the simulated characters and did not change during the walking task. Note that for each trial, different nearby were chosen. This was measured in meters.

### 3.5 Procedure

The experiment was conducted at the motion capture studio of our department. Once the participants arrived, the research team provided information to them about the project, and they were asked to read the provided consent form, which was in accordance with the Institutional Review Board of Anonimus University, and sign it in case they agree to participate in the study. Upon agreement, the participants were asked to complete a demographic questionnaire. In the next step, the researcher helped the participants with the backpack computer and the head-mounted display. Once everything was set, the participants were asked to take a short walk within a virtual replica of the motion capture studio to ensure they were comfortable enough when wearing all the devices.

After becoming comfortable and familiar with the virtual reality equipment, the researchers asked the participants to remove the headset and move toward a marked location in the real environment and face the opposite direction. Once participants landed on the marked position and before the experiment started, the researcher informed them that once the application started, they would be placed into a virtual metropolitan city and the task they would have to perform was to cross the road. The participants were informed that at first, the traffic light would turn green and then they would hear a "beep" sound that would signal them when they should start walking. It was decided to delay the time in which the participants should start walking to ensure that a proper number of characters surrounded the participants. This helped place the participant within the virtual crowd. Participants were told they could have breaks between the trials of the conditions if needed and that they had full permission to leave at any time.

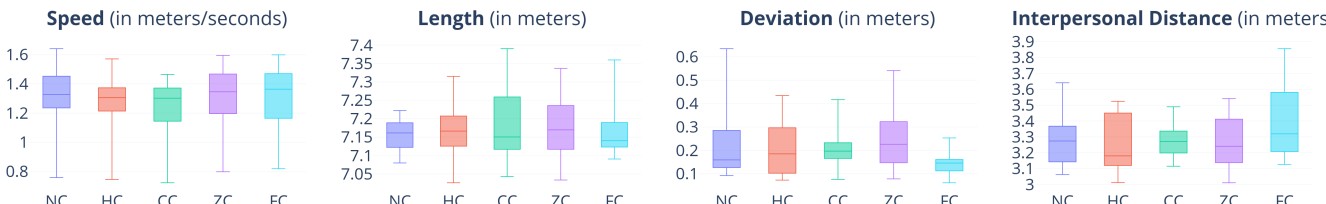

Figure 4: Boxplots of the obtained results for each measurement. NC: neutral crowd, HC: human crowd, CC: cartoon crowd, ZC: zombie crowd, and FC: fantasy crowd.

All participants were informed that they would cross the virtual crosswalk to reach the opposing sidewalk 20 times (5 conditions × 4 trials). They were also told they would be informed when the experiment had ended. Once the participant finished each trial, they were asked to remove the head-mounted display and move back to the marked location on the other side of the room. This process is repeated for all repetitions. Participants were informed by the research team once the experiment was completed. The balance for first-order carry-over (residual) effects between the conditions was ensured using Latin squares [18]. The total duration of the experiment lasted on average 30 minutes.

## 4 RESULTS

This section presents the results obtained from the main study. All the analyses were performed using IBM SPSS v. 23.0 [24] software. One-way repeated measure analysis of variance (ANOVA) was used to analyze the obtained data using the five experimental conditions as independent variables and the motion measurements as dependent variables. The normality assumption of the measurements was evaluated graphically using Q-Q plots of the residuals [12]. The Q-Q plots indicated that the obtained data fulfilled the normality assumption. The individual differences were assessed using a post hoc Bonferroni test if the ANOVA was significant. A $p < .05$ was deemed as statistically significant.

The effect of crowd characters on the participants' movement behavior was compared using four objective measurements (speed, deviation, length, and interpersonal distance) across the five experimental conditions (NC, HC, CC, ZC, and FC). Boxplots of the results are presented in Figure 4 and descriptive statistics for the results in Table 1. No significant differences were found across the five experimental conditions for the **length** measurement [$\Lambda = .249$, $F(4, 14) = 1.160$, $p = .370$, $\eta_p^2 = .249$].

In relation to the **speed** that participants needed to cross the virtual crosswalk, significant results were found [$\Lambda = .456$, $F(4, 14) = 4.019$, $p < .05$, $\eta_p^2 = .535$] across the five experimental conditions. Pairwise comparisons indicated that the mean speed for the CC condition was significantly lower than the FC condition at the .005 level. No other difference was found across the examined condition.

The **deviation** of participants was also significant across the five experimental conditions [$\Lambda = .466$, $F(4, 14) = 4.018$, $p < .05$, $\eta_p^2 = .534$]. Pairwise comparison indicated that the mean deviation for the FC condition was significantly lower than that for the CC condition at the $p < .05$ level, and the ZC condition at the $p < .05$ level. No other difference was found across the examined condition.

Finally, in relation to the average **interpersonal distance**, significant results were found across the five experimental conditions [$\Lambda = .440$, $F(4, 14) = 4.456$, $p < .05$, $\eta_p^2 = .560$]. Pairwise comparison indicated that the mean interpersonal distance during the FC condition was significantly higher than that for the HC condition at the $p < .005$ level, and the ZC condition at the $p < .05$ level. No other differences across conditions were found.

Table 1: Descriptive statistics (Mean [M], Standard Deviation [SD], Minimum [Min] and Maximum [Max] value) for each measure across experimental conditions ($N = 18$), and patterns of differences.

| Condition | M | SD | Min | Max | Results |
|---|---|---|---|---|---|
| **Speed** | | | | | |
| NC | 1.29 | .18 | .76 | 1.53 | CC<FC |
| HC | 1.26 | .17 | .74 | 1.57 | NC=HC=CC=ZC |
| CC | 1.23 | .18 | .72 | 1.43 | NC=HC=ZX=FC |
| ZC | 1.29 | .18 | .80 | 1.54 | |
| FC | 1.31 | .19 | .82 | 1.58 | |
| **Length** | | | | | |
| NC | 7.15 | .04 | 7.08 | 7.22 | NC=HC=CC=ZC=FC |
| HC | 7.17 | .07 | 7.02 | 7.31 | |
| CC | 7.18 | .09 | 7.04 | 7.39 | |
| ZC | 7.17 | .07 | 7.03 | 7.33 | |
| FC | 7.16 | .06 | 7.09 | 7.35 | |
| **Deviation** | | | | | |
| NC | .23 | .15 | .09 | .64 | FC<(CC=ZC) |
| HC | .21 | .11 | .07 | .43 | NC=HC=CC=ZC |
| CC | .20 | .08 | .08 | .42 | NC=HC=FC |
| ZC | .25 | .14 | .08 | .54 | |
| FC | .15 | .05 | .06 | .25 | |
| **Interpersonal Distance** | | | | | |
| NC | 3.28 | .15 | 3.06 | 3.64 | (HC=ZC)<FC |
| HC | 3.25 | .17 | 3.01 | 3.52 | NC=HC=CC=ZC |
| CC | 3.29 | .11 | 3.11 | 3.49 | NC=HC=FC |
| ZC | 3.28 | .16 | 3.01 | 3.54 | |
| FC | 3.38 | .23 | 3.13 | 3.85 | |

## 5 DISCUSSION

The study was conducted to understand the effects of virtual characters' appearances that belong to a moving virtual crowd on human movement behavior. The participants were immersed in a virtual metropolitan city surrounded by a virtual crowd population that was moving towards the opposite sidewalk and they were instructed to walk toward that direction. No additional instructions on moving within the virtual environment were given to participants. The collected data was analyzed to determine whether the participants' movement behavior affected by the virtual crowd and more specifically, by the appearance of the virtual characters that composed the examined crowd condition.

By analyzing the collected data, it was discovered that the speed, deviation, and interpersonal distance variables were altered when participants walked in the different crowd conditions. Participants' speed was significantly lower when exposed to a virtual crowd composed of cartoon characters compared to a virtual crowd composed of fantasy characters. We also found that participants' deviation during the crowd condition composed of fantasy virtual characters was significantly lower than that for the crowd composed of cartoon and

zombie characters. Finally, participants' interpersonal distance to the four closest characters in the forward direction was significantly higher during the fantasy crowd condition compared to human and zombie crowd conditions.

An interpretation of slower speed could be that cartoon characters felt less human-like and eerier. Participants could have taken more time to examine the cartoon characters. On the other hand, participants could have easily recognized fantasy characters as something they are familiar with in video games. They were more comfortable with walking with fantasy characters as opposed to cartoon characters. Hence they were able to accomplish their task faster with fantasy characters.

A similar interpretation can be applied to why their deviation from fantasy characters is low in comparison to cartoon characters. They deviated more as they were unsure of the intentions of the cartoon characters. The participants probably recognized zombie characters as threatening and were apprehensive when walking close to them which could explain why the deviation with zombie characters is more in comparison to fantasy characters.

Fantasy characters had the most interpersonal distance in comparison to zombies and humans. This does seem contradictory to the previous results. An explanation could be that, even though the interpersonal distance was higher, participants easily recognized it as the most comfortable distance they should keep from the characters. Interpersonal distance with the other characters could be more sporadic, which leads to it being less as an average. This also explains why the deviation was less and the speed was more with fantasy characters.

In regards to the speed of the crowd, one interpretation is that because the cartoon characters could be considered to be more cute and approachable than the zombie or fantasy characters they decided they could walk slower. This may be because participants felt more at ease with the simpler character designs. By similar metrics, an explanation for the difference in interpersonal distance is that the participants felt more natural being closer to the human-like characters and might have been wary of the zombie and fantasy characters so they kept their distance.

The deviation in speed could be attributed to the fantasy characters sharing similar physical proportions and body structure to real humans, which could be interpreted by a participant as being included in a crowd that more closely mirrors a typical crowd of humans, albeit in unusual attire. This, in turn, may have caused the participants to become more cautious while moving with the cartoon characters, due to them being constantly aware of the disproportionated body features, and accidentally moving into the personal space of (or in this case, colliding with) the virtual cartoon characters. Similar interpretations can be made when comparing the cartoon and zombie crowds, with the fantasy crowd in regards to the deviation of the participant. However, the results for the interpersonal distance suggest something very interesting. Even though participants' behaviors may imply that they were more comfortable moving with the fantasy characters, in comparison to the zombies and cartoon characters, the results show they kept the most distance between themselves and the fantasy characters. This could indicate that the ability to identify characters as humans could be the root cause of the differences here. The cartoon characters have slightly different proportions and the zombies are unmistakably not human while the fantasy characters are human with non-standard attire.

Unfortunately, we were not able to see any significant differences in the trajectory length of the participants' path. The total distance was included to confirm whether or not the global deviation was significant, the various characters were not triggering avoidance mechanisms in the participants where its statistical significance was found. While anecdotally there were a few instances of participants moving out of the way in both zombie and fantasy characters, no statistical significance was found. This could be for a number of

reasons, but we find the most likely case to be the objective of walking to the other side of the road to be more enticing that the characters surrounding it.

A few limitations should be noted in this study. The lack of including a questionnaire that includes the participants' presence, embodiment, and trust would have been helpful additional information for analysis but would have taken too long to implement. The extent of a participant's immersion may have assisted in understanding their movement patterns. Additionally, the characters walked at a consistent speed which may have had an effect on the participants' perception of realism. There may also be a potential issue with the proportions of the cartoon characters used. While all models had roughly the same heights and shoulder widths, there are some minor variations in the head size and limb proportions of a few cartoon characters. The methods used to calculate the interpersonal distances will not be affected by these minor variations but the participant's perception of proximity to the characters may have been affected. This could potentially lead to the participants to change either their trajectory or movement speed slightly.

## 6 CONCLUSIONS AND FUTURE WORKS

From the measurement taken it was found that the appearance of virtual characters that belong to a virtual crowd could be a factor to alter the movement behavior of participants. Specifically, the speed, deviation, and interpersonal distance were discovered to have significant differences between the five experimental conditions. It is shown that different types of avatars will affect the distance they keep in a crowd, their overall speed, and the deviation from the global crowd trajectory.

There are numerous avenues in which to investigate virtual crowds and the effects they have on participants that extend beyond changes in appearance. One such way is to change various locomotive aspects of the crowd including the direction to be towards the participant, curving the angle of the path that the crowd should follow, or setting the crowd to a pattern like Perlin noise or flocking behavior. It would also be possible to extend these studies to other crowd population interactions (walking alone or in groups, stopping at storefronts, talking, waiting, etc.) and understand the way the participants interact with such behaviors and understand the virtual crowd in general.

While this study has a limited scope on the role of characters on human locomotion behaviors in virtual reality crowds we believe this may provide useful insights for future researchers on similar lines of inquiry. Researchers and developers or virtual reality games and applications can utilize this information to better predict the implications of designs that include virtual reality crowds.

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
