# OpenReview forum: "Walking within a Crowd Full of Virtual Characters: Effects of Virtual Character Appearance on Human Movement Behavior"
_graphicsinterface.org/Graphics_Interface/2020/Conference — Submitted to GI 2020_

### Official Review · AnonReviewer2 · 2019-12-29
**Manuscript could be motivated stronger**

**Confidence:** 3
**Rating:** 4

**Review:**


This manuscript considers the question of whether people in room-scale VR walk differently in crowds depending on the different types of avatars in the VR environment. The authors approach this problem by designing an experiment where participants don head-mounted VR displays and physically walk about 8m while viewing a simulated crosswalk (with other avatars). The authors find that based on the type of avatar, there are some differences in people's walking speeds, and some other variables (interpersonal) distance that the authors are measuring.

This manuscript is straightforward, but I think the main challenge for me is that the authors have not done a good enough job of motivating the particular work. Why are we concerned specifically with the avatars? What does the prior work tell us about this that makes it important to study this problem? And, what hypotheses should we have about these differences a priori? I think right now, the explanations and interepretations feel very ad hoc, speculative, and unsubstantiated. It also means that it's difficult to necessarily be confident extrapolating from the results. The other challenge is that, even though the authors have taken considerable time explaining the experimental setup, there are still points of uncertainty. While the clarity issues may seem pedantic, I think for me, they stem from the uncertainty about *why* we might see results. For a piece of research the reports on a study, it is important to be able to replicate the study set up for reproducibility and verification. I do not think the manuscript is ready for publication.

What I like about the manuscript is that the research question is straightforward (even if not well-motivated), and so the study design is also fairly straightforward. While I do not agree with the choices the authors made in the measures they use, these are understandable choices. So in this sense, the manuscript is easy to understand at a high level.

Below, I detail some of the more substantive issues that I think the authors should consider moving forward:

* The authors need to provide a clearer explanation of the motivation: why is answering this research question important or valuable? Has prior work suggested that this is something important or valuable to study?
* The related work is extensive, but there isn't a really great narrative that brings us through it. It mostly feels like a laundry list. Suggestion: for each paragraph, what is the message that is being conveyed? What's the purpose of the paragraph? How does it help or set up the context for the present work? is it related? is it peripheral?
* While the setup of the study is straightforward, it isn't clear why we should expect there to be differences between these avatar types a priori.
* To me, Figure 2 would make more sense to show the backs of these characters, I think, since this is what participants were looking at

Within the study:

* Clearer description of the participants would be valuable. Have these people had experience with VR? Have they had experience with room-scale VR?
* Do the virtual characters move in a straight line? Do they disappear when they reach the other side? Or, does the crowd continue to accumulate? Based on the manuscript, I think each character moves at a different rate; do they vary in their own individual movement speed (i.e. variance/deviation), or is their individual movement rate constant?
* Did the characters similarly avoid the participant?
* How much delay time was there before participants were allowed to start? How was the start signaled?
* On the metrics:
	* "Speed would tell us if the participants either felt more at ease or on edge in their walk," -- why would you say this? I'm not sure if I agree with this.
	* " Deviation: The average deviation (absolute value) between the global trajectory of the virtual crowd and the trajectory of the participant. The average deviation was measured in meters." -- Is this a speed measure? Or a distance measure? Where is the trajectory measured? Wouldn't this be angular difference?
	* I'm not sure I understand the "length" measure. Is this the *distance traveled* by the participant? How is this measured? How far do participants travel in the virtual world if they were going in a straight line?
* There are statistical differences between these measures, but the explanations feel speculative. It might have been valuable to inform these by including some qualitative reports from participants. Did participants have explanations for this? Did participants spend time looking at the avatars? Maybe some qualitative description of what they did would be useful. Did participants avoid the characters? Did they just barrel through them? And, so on.
	* The 3m interpersonal distance seems a lot, given that the characters held a distance of 0.76cm min ... how far apart were characters from one another?

---

### Official Review · AnonReviewer1 · 2020-01-04
**Overall well written, but issues with analysis and motivation.**

**Confidence:** 4
**Rating:** 5

**Review:**

This paper presents a study analyzing participants walking behaviour amongst five different types of virtual crowds in VR. They find differences in participants behaviours depending on which type of virtual character the participants were walking alongside.

I don't believe this paper is ready for publication in its current form, though it might be if two concerns can be addressed: The analysis and presentation of results, and the motivation and relevant use cases that these findings might have. Additionally, I think the effects of the findings are actually pretty small, but if the paper can be better motivated than perhaps the small findings may be valuable.

Regarding the analysis, there are several errors that should be corrected, and could potentially be without impacting the findings of the study. First, a one-way ANOVA was used when an RM-ANOVA should have been used. Participants performed all five conditions, so the samples and data should be treated as related samples. Secondly, the paper states that a bonferroni test was used, but that is a *correction* not a test. The Bonferroni correction should be *applied* to a test, such as a paired samples t-test. Additionally, meaningful effect sizes should be presented. Partial eta-squared is difficult to interpret, and something like Cohen's d, or the natural effects (e.g., in m or m/s) should be presented to let the reader know the magnitude of these effects.

I think the magnitude is actually pretty small. There are only slight differences in the means for many of the measures, and with 5 factors, multiple metrics, and 18 participants running 25 trials each, the probability of some significant factors is pretty high.

Regarding motivation, I just don't understand what these results add. Are they actionable for any use case (current, or imagined?), or do they tell us something interesting about human psychology? The discussion presents some potential explanations as to the results, but I'm still left wondering what to do with this information now that I have it.


I have no expertise in understanding crowd simulation.

---

### Official Review · AnonReviewer3 · 2020-01-09
**Interesting research topic, lacking in underlying theory and explanation of results**

**Confidence:** 3
**Rating:** 4

**Review:**

In this submission, the authors investigate the influence of the visual appearance of crowd members on people's traversal of said crowd in a virtual reality environment. The crowd appearance (IV) varies between neutral, human, cartoon, zombie, and fantasy; the authors measure people's traversal (DV) in terms of walk speed, path length, deviation from from average path, and average distance between participant and (virtual) crowd characters. The authors reported multiple significant differences, mostly between fantasy and cartoon, such that, broadly speaking, participants traversed cartoon crowds slower than fantasy crowds. The authors presented multiple, in parts contradictory, explanations for this result (e.g., cuteness and eeriness).

Overall, I find this research topic could be of interest to the HCI community, especially CHIPLAY. As a minor note, however: I wish the authors would have motivated their research with some potential real-world applications; just because something is an "overlooked factor" does not necessarily make it an relevant research topic.

A more severe issue for me is the lack of a theoretical foundation for the the expected results of the experiment. It seems to me that the authors simply ran a study in hope of finding some effect, without much a priori hypothesizing of possible outcomes. This sets of the usual chain of doubt about the validity of the results: finding some significant differences between conditions, struggling to explain some of the results, skipping over an analyses of effect sizes, and uncertainty about the reproducibility of the results. This becomes evident in the discussion section, where the authors struggle finding concise explanations for the experiment results. Almost all explanations sound contrived to me, and could be easily reversed if the data would have been different (e.g., cartoons are usually simplified abstractions, so people need less time to visually analyze them compared to complex fantasy characters).

While I find the research idea compelling, I ultimately cannot argue for accepting this paper as it does not explain the results in a satisfactory way: even if the results are reproducible, how can other researchers or practitioners generalize them and apply to their work?

---

### Meta-Review · Area_Chair1 · 2020-01-09

**Recommendation:** Reject
**Confidence:** 5

**Metareview:**

Based on the reviews, the reviewers are unanimous that the manuscript is not quite ready to be published.

In general, while the reviewers thought the idea was interesting and straightforward, they found the work to be poorly motivated.

I'd recommend the authors examine these reviews for more details.

---

### Decision · Program_Chairs · 2020-01-11

Reject